# Smart Anklet Use to Measure Vascular Health Benefits of Preventive Intervention in a Nature-Based Environment—A Pilot Study

**DOI:** 10.3390/ijerph21050605

**Published:** 2024-05-09

**Authors:** Maja B. Stosic, Jelena Kaljevic, Bojan Nikolic, Marko Tanaskovic, Aleksandar Kolarov

**Affiliations:** 1Faculty for Health and Business Studies, Singidunum University, 14000 Valjevo, Serbia; jkaljevic@singidunum.ac.rs (J.K.); bojannikolic@singidunum.ac.rs (B.N.); 2Institute of Public Health of Serbia “Dr Milan Jovanovic Batut”, Department for HIV, Hepatitis, STIs and Tuberculosis, 11000 Belgrade, Serbia; 3Faculty for Technical Science, Singidunum University, 11000 Belgrade, Serbia; mtanaskovic@singidunum.ac.rs; 4New Jersey Institute of Technology, Electrical and Computer Engineering Center, Ewing, NJ 07102, USA; kolarov@yahoo.com

**Keywords:** vascular health, nature-based intervention, smart anklet, prevention, peripheral arterial occlusive disease

## Abstract

The present study aimed to investigate the associations between nature-based intervention and peripheral pulse characteristics of patients with PAOD using new smart technology specifically designed for this purpose. A longitudinal panel study performed between 1 January 2022 and 31 December 2022 included 32 patients diagnosed with peripheral arterial occlusive disease (PAOD) who were treated in the vascular surgeons’ hospital “Dobb” in Valjevo. These patients were exposed for six months to moderate-intensity physical activity (MPA) in a nature-based environment. They practiced 150 to 300 min of walking 6 km/h and cycling activities (16–20 km/h) weekly as recommended for patients with chronic conditions and those living with disability. Univariate logistic regression analysis was used to identify factors associated with major improvements in peripheral pulse characteristics of patients with PAOD. After six months of MPA, half of the patients (50%, 16/32) achieved minor, and half of them major improvements in peripheral pulse characteristics. The major improvements were associated with current smoking (OR = 9.53; 95%CI = 1.85–49.20), diabetes (OR = 4.84; 95%CI = 1.09–21.58) and cardiac failure, and concurrent pulmonary disease and diabetes (OR = 2.03; 95%CI = 1.01–4.11). Our pilot study showed that patients with PAOD along with other chronic conditions and risk factors benefited more from continuous physical activity in a nature-based environment.

## 1. Introduction

Vascular diseases (VD) are the leading causes of morbidity and mortality worldwide, presenting as illnesses of the large and small blood vessels, including lower limb peripheral arterial occlusive disease (PAOD). The global numbers of prevalent cases and deaths due to PAOD have risen consistently each year since 1990, resulting in a 2-fold increase to 113 million cases and 74,100 deaths in 2019 [1,2]. In recent years, an average of 55% of people who died in Serbia were victims of cardiovascular diseases. In relation to all causes of death, during 2019, in Serbia, 24,362 men (45.4%) and 29,306 women (54.6%) died of diseases of the heart and blood vessels. The average death rate from diseases of the heart and blood vessels in Serbia in the period from 2013 to 2019 was 754.2 per 100,000 inhabitants, and the trend is increasing [3]. The modern urban lifestyle is associated with chronic stress, a high prevalence of smoking, insufficient physical activity, and other risk factors for VD. Access to nature may produce a wide variety of health benefits [4]. Positive associations between exposure to nature, higher levels of physical activity, and lower levels of cardiovascular disease were noticed in cross-sectional observational studies [5,6]. Due to health sensors, smart home devices, and the internet of medical things available today, new opportunities have opened to effectively manage different health conditions with personalized screening and timely intervention [7]. These fast-growing, low-cost, and widely available resources can help predict one’s risk for different conditions including PAOD, saving limbs and lives [8]. Several platforms have been designed that enable monitoring of cardiovascular health where the data are sent to a central server where it is processed for the purpose of notifying the patients or social or health care workers of the overall health of the patient. One of the examples is the Body Cardio Scale (Nokia, Helsinki, Finland) that can assess cardiovascular health by heart rate and pulse wave velocity; it has been shown that measuring pulse wave velocity could assist in early identification of PAOD [9]. However, the effectiveness of these home-based technologies to determine PAOD or provide clinically meaningful information about cardiovascular health needs to be clinically validated in future studies [10]. In recent decades, optical sensor technology and the re-establishment of photo plethysmography as a non-invasive physiological measurement method were used for PAOD assessment [11,12]. Although smart devices and sensors are widely used in medical practice as efficient, patient friendly, and objective monitoring of medical conditions, links between their use for nature-based intervention and peripheral vascular diseases are not investigated. The aim of the study was to investigate the associations between nature-based intervention and peripheral pulse characteristics of patients with PAOD using a new smart device specifically designed for this purpose.

## 2. Materials and Methods

### 2.1. Study Design

We performed a longitudinal panel study from 1 January 2022 to 31 December 2022 at the vascular surgeons’ hospital “Dobb” in Valjevo, Serbia. The first cross-section of the study participants was conducted before nature-based intervention, while the second was conducted six months later.

### 2.2. Study Participants

Criteria for inclusion in the study were patients aged 18 and above who were diagnosed with and treated for lower limb PAOD, were able to perform moderate-intensity physical activity, and who agreed to participate in the study. Individuals with mental inability to understand the goals and the procedures of the study, as well as the ones who refused to participate, were not included.

The initial diagnosis of PAOD is based on physical examination and duplex ultrasonography. The patients included in the study regularly used their medications. The study had no influence on therapy.

### 2.3. Data Collection

#### 2.3.1. Measures

The peripheral pulse characteristics for the study purposes were measured by a specifically designed device called a smart anklet (Figure 1).

After the initial assessment carried out by three measurements (in the morning, in the afternoon, and in the evening), patients were exposed for six months to moderate-intensity physical activity (MPA) in a nature-based environment. They practiced 150 to 300 min of walking 6 km/h and cycling activities (16–20 km/h) weekly as recommended for patients with chronic conditions and those living with disability.

The second assessment was performed after six months of physical activity; three measurements were conducted in the same way as the initial assessment.

Variables monitored before and after preventive intervention were: Pain when walking, self-reported by the participants;Characteristics from the measurements by the smart anklet device: claudication distance defined as the distance after which the patient is forced to stop because of severe pain and muscle cramps, wave amplitude—dampened, wave amplitude—delayed, and wave amplitude—reduced.

The average values were calculated for the initial assessment and compared to the average values of the measurements within the second assessment.

In addition, patients were classified by the Fontaine stage before and after the preventive intervention. The Fontaine classification describes four stages in the clinical presentation of lower extremity arterial disease: 

Stage I—Patients who are asymptomatic most of the time, but in whom a careful history may reveal non-specific, subtle symptoms, such as paresthesia. Physical examination may reveal cold extremities, reduced peripheral pulse, or murmurs in the peripheral arteries.

Stage II—Intermittent claudication. Patients usually have a consistent distance at which the pain appears:

Stage IIa—Intermittent claudication after more than 200 m of walking.

Stage IIb—Intermittent claudication after less than 200 m of walking.

Stage III—Rest pain. Rest pain appears especially during the night when the legs are raised up on the bed, which diminishes the gravitational effect present by day; also, during the night, the lack of sensory stimuli allows patients to focus on their legs.

Stage IV—Ischemic ulcers or gangrene (which may be dry or humid) [13].

To assess the outcomes of the preventive interventions, the improvement scores were calculated for the following variables: pain when walking, claudication distance defined as the distance after which the patient is forced to stop because of severe pain and muscle cramps, wave amplitude—dampened, wave amplitude—delayed, and wave amplitude—reduced. If the improvement is presented, the variable is marked with 1; if not, it is marked with 0. The highest score was 5 and the lowest was 0.

Based on the median distribution of scores, the improvements were classified as minor and major. The improvement is classified as minor if 1 or 2 variables improved, while it is classified as major if 3 or more variables improved during the study period.

Results were presented as frequency (percent), median (range), and mean ± standard deviation (SD). Univariate logistic regression analysis was used as the method for analyzing binary outcomes and potential predictors. Due to the small number of study participants, it was not feasible to design a multivariate logistic regression (MLRA) model. All *p*-values less than 0.05 were considered statistically significant. Statistical data analysis was performed using IBM SPSS Statistics 22 (IBM Corporation, Armonk, NY, USA).

#### 2.3.2. Instrument

The detection of the arterial flow by the sensors within the device was based on the principle of photo plethysmography (PPG). PPG sensors used a green light-emitting diode (LED) as the main light source. The PPG device contains a light source and a photodetector. The light source emits light into the tissue while the photodetector measures the reflected light from the tissue. The reflected light is proportional to variations in blood volume. Changes in blood volume are measured (calculated) based on the amount of light detected. Wave amplitude corresponds to the blood volume. By measuring the difference in two measurements, we make conclusions as to whether the wave amplitude is dampened, delayed, or reduced.

The hardware device consists of a specifically designed printed circuit board (PCB) and suitable storage that houses the processor and memory of the device with a strip for easy mounting. A key element in the PCB design was the PPG sensor. In particular, a ready-made solution with a unique integrated electro component that contains an LED and a corresponding photo receiver in one unique case was used (Available online: https://valencell.com/ppgsensors/, accessed on 1 May 2024). The electronic board is powered from a small 5 V watch battery, where the selection of the battery was made so that the device can work for a whole day without interruption. For the needs of the longest possible autonomy of the device, the design of the printed circuit board, consumption optimization methods were used. The board can communicate with the outside world via the USB bus, through which it is also possible to charge the battery. There is a special part of the electronic system designed for monitoring the state of the battery and for charging via USB port. The device also has a diode to indicate battery status to the outside world. The printed circuit board also contains a simple mechanical button for pain indication, that the patient presses when they feel severe pain in their legs. It is connected to the rest of the circuit and has a clear indication signal. The device is controlled by a microcontroller. This controller has the role of counting the patient’s steps from the accelerometer data. The board also contains additional flash memory. The flash memory is used to record signals from the PPG sensor, the number of steps, and recorded pressures on the pain indication button.

Special communication protocol is implemented on the microcontroller that allows for simple reading of the data from the flash memory via the USB bus.

For better adhesion to the patient’s skin and adaptability to the shape legs, the PCB was made in elastic printed technology, which allows it to bend. The whole design is packed in a very small area so that the entire device is as least robust and noticeable as possible. The PCB relates to straps for bindings that were used for simple clamping. The software used for visualization and data processing from the device to a PC is a custom-made software written in Python programming language, with capabilities for graphical visualization of the recorded data. The changes in blood pressure are not measured directly, but are estimated from the light-intensity changes using a sophisticated algorithm. Additionally, the number of steps and claudication distance is not measured directly, but is estimated based on the measurements from accelerometers. The smart anklet device with sensors was placed under the joint of the leg, i.e., the inner and outer malleolus. 

In addition to the data provided from the instrument, other data included patients’ medical records containing data related to socio-demographic characteristics, habits and behaviors, and concomitant diseases. 

### 2.4. Ethical Approval

The Ethics Committee of the University of Singidunum (agreement number I-151-1/2021 issued on 30 December 2021), Faculty of Health and Business Studies, approved the study. Personal identifiers of study participants were coded, and patient records were anonymized and de-identified prior to analysis to maintain confidentiality. All participants signed a voluntary informed consent form, providing demographic and clinical data and participation consent.

## 3. Results

Out of 32 study participants, 28 (87.5%) were male. The dominant age category was 65+ years, with 27 (84.4%) patients in that age range. Most participants were married, 21 (65.6%), 18 of them had completed university education (56.2%), and 17 of them (53.1%) were retired. They lived predominantly in urban areas (20 participants (62.5%)). Less than half were smokers (43.8%), while more than half of the study participants used alcohol (53.1%) at the time of the study. One fourth of them had chronic obstructive pulmonary disease (COPD) and asthma, one half had diabetes, and 6.3% had thyroid disease. More than one fourth of them (28.1%) had one co-morbidity, 25% had two, 25% had three, and 21.9% had four co-morbidities. 

Statistically significant differences were observed in walking pain, claudication distance (m), wave amplitude—dampened, wave amplitude—delayed, and wave amplitude—reduced, before and after the prevention intervention (Table 1), as well as in Fontaine stages IIa and IIb before and after the preventive intervention (Table 2).

Out of the five variables monitored before and after preventive intervention, improvements in two variables were recorded in 13 patients (40.3%), while improvements in five variables were noted in only 2 (6.3%), as presented in Table 3.

After six months of MPA, half of the patients (50%)achieved minor improvements, and half of them major improvements in the observed variables. Major improvements were statistically significantly associated with university education (*p* = 0.006), current smoking (*p* = 0.002), and having diabetes (*p* = 0.034) and comorbidities (*p* = 0.045), as described in Table 4.

In univariate logistic regression analysis, factors significantly associated with major improvements were as follows: currently smoking (OR = 9.53; 95%CI = 1.85–49.20), diabetes (OR = 4.84; 95%CI = 1.09–21.58) and cardiac failure and concurrent pulmonary disease and diabetes (OR = 2.03; 95%CI = 1.01–4.11)—Table 5.

## 4. Discussion

To the best of our knowledge, there were no studies investigating the associations between physical activity in a natural environment and peripheral pulse characteristics of the patients with PAOD. Moreover, there were no studies investigating medical conditions using smart anklet devices. After six months of performing moderate-intensity physical activity in a nature-based environment, we found that half of the study participants achieved minor improvements, while half of them achieved major improvements in the peripheral pulse characteristics measured by the smart anklet. In line with our results, studies performed in Italy [14,15,16] showed that a structured exercise program resulted in progressive improvement in vascular function and the ability to walk, a lower death rate, and better long-term clinical outcomes, particularly among women, and participants who attained a moderate increase in exercise capacity. Studies from the USA [17,18] demonstrated the effects of exercise rehabilitation in older patients with peripheral arterial occlusive disease and symptomatic and functional improvement in performance parameters. The same study showed potential possibility in maintaining these benefits and improve health-related quality of life with the continuation of the exercises. Major improvements in the peripheral pulse parameters among our study participants, during the physical activity in a nature-based environment, were associated with current smoking. The improvement was larger for those who smoked in comparison to those that did not smoke. A co-occurrence of smoking and physical inactivity is reported by many studies [19,20], as well as a high percentage of PAOD attributable to smoking [21]. Taking all of this into account, it is not surprising that behavioral change in terms of continuous moderate-intensity physical activity had a favorable outcome on the pulse characteristics, either directly, by progressively increasing blood flow, or indirectly, by continuous reduction in the number of cigarettes smoked during exercises [22]. Also, our results showed that moderate-intensity physical activity in a nature-based environment among patients with diabetes was associated with major improvements in the peripheral pulse parameters. Exercise, when performed on a regular basis, is a well-accepted strategy used to improve vascular function in patients with diabetes [22]. Current guidelines for patients with type 2 diabetes largely advocate the importance of physical activity as a therapeutic tool to improve and control several cardiovascular risk factors including glycemic control, blood pressure, and endothelium function. In line with these findings, the American Diabetes Association (ADA) states that patients with type 2 diabetes should accumulate at least 150 min of moderate-intensity physical activity per week to maintain or improve health [23,24]. Although walking is an important component in the management of PAOD as it improves lower limb function, one challenge reported by clinicians when recommending walking therapy for patients with PAOD is the motivation to adhere to this behavior [25]. One of the motivation factors for our patients to sustain regularly physical activity might be a natural environment, as it has been found that natural environments and accessible green spaces have both direct and indirect influence on health and wellbeing [26]. Natural environments and approachable green and blue spaces support active recreation, relaxation, and consolidation from daily stress [27]. Since our study was performed during the COVID-19 pandemic, the possibility of spending time in a natural environment might be something that patients are particularly lacking, and this issue might increase motivation as well. Furthermore, we found that cardiac insufficiency, pulmonary diseases, and concurrent diabetes, among PAOD patients who performed moderate-intensity physical activity in a nature-based environment, were associated with major improvements in the peripheral pulse parameters. As our study participants were predominantly above 65 years of age, the ageing phenomenon has led to a substantial increase in chronic conditions, consequently resulting in the rising prevalence of multi-morbidity, which is most commonly described as the presence of two or more long-term conditions [28,29,30]. The management of multi-morbidity is a complex process, and has recently become an emerging public health priority for professionals and health care systems [31,32,33]. Although physical activity has been recommended as one of the main lifestyle behaviors in the management of several chronic conditions, more often lack of physical activity is reported among the patients particularly in low- and middle-income countries [34,35]. Multi-center study of Vancampfort D. et al. performed in 46 low- and middle-income countries, showed that chronic conditions were associated with low physical activity most notable among the older population, due to mobility difficulties, pain, depression and sleep problems [35]. Other studies reported lack of time, social support, energy, motivation, skill, resources, and fear of injury during practice, as predictors of the perception of barriers to physical activity [35,36,37]. For all of the reasons mentioned above, in our patients, good results with continuous physical activity might be achieved due to using the natural environment as a place for performing physical activity. 

There are severallimitations of the study. The main limitationis the absence of a control group. Due to limitations in cost and time, and the inability of certain number of patients with PAOD to participate in the study due to their objective inability to perform continuous physical activity, we included a limited number of respondents. It resulted in non-equal distribution of patients between the age groups. In addition, we measured only several peripheral pulse characteristics. Therefore, a larger sample size is needed to explore further associations and increase the strength of the study. Despite limitations, our study results provided initial associations between independent and dependent variables and the possibilities of using smart devices for measuring peripheral pulse characteristics. These findings should be further explored in future research.

## 5. Conclusions

Our pilot study showed that smart anklets can be used as portable devices to measure several basic peripheral pulse characteristics in patients with PAOD. In addition, patients with PAOD along with other chronic conditions and risk factors benefited more from continuous physical activity in a nature-based environment. Further studies with a larger sample size are needed to explore independent predictors for improvement in peripheral pulse characteristics and other possibilities for future developments of smart anklets.

## Figures and Tables

**Figure 1 ijerph-21-00605-f001:**
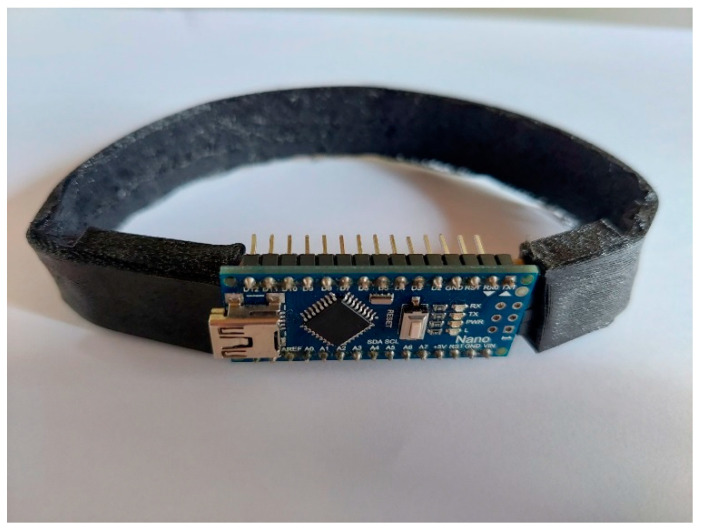
Smart anklet device.

**Table 1 ijerph-21-00605-t001:** Distribution of variables monitored before and after the preventive intervention.

Variable	Before *n* (%)	After *n* (%)	*p* Value
Walking pain Yes No	23 (71.9) 9 (28.1)	6 (18.8) 26 (81.2)	0.001
Claudication distance (m) 50–100 100–200 200–	11 (34.4) 17 (53.1) 4 (12.5)	2 (6.3) 13 (40.6) 17 (53.1)	0.001
Wave amplitude—dampened Yes No	19 (59.4) 13 (40.6)	5 (15.6) 27 (84.4)	0.001
Wave amplitude—delayed Yes No	19 (59.4) 13 (40.6)	7 (21.9) 25 (78.1)	0.002
Wave amplitude—reduced Yes No	19 (59.4) 13 (40.6)	4 (12.5) 28 (87.5)	0.001

**Table 2 ijerph-21-00605-t002:** Fontaine stage of patients before and after the preventive intervention.

**Fontaine Stage**	**Before Intervention**		**After Intervention**		***p* Value**
	** *n* **	**%**	** *n* **	**%**	0.001
IIa	4	12.5	17	53.1	
IIb	28	87.5	15	46.9	

**Table 3 ijerph-21-00605-t003:** Number of variables improved after preventive intervention.

*N* of Variables Improved	*N* (%) of Participants
1	3 (9.4)
2	13 (40.3)
3	3 (9.4)
4	11 (34.3)
5	2 (6.3)

**Table 4 ijerph-21-00605-t004:** Distribution of improvements in observed variables among the study participants (*n* = 32).

Variable	Minor Improvement *n* (%)	Major Improvement *n* (%)	*p* Value
Sex Male Female	13 (81.3) 3 (18.7)	15 (93.8) 1 (6.3)	0. 600
Age (mean ± SD)	62.06 ± 7.9	59.06 ± 6.5	0.252
Marital status Married/cohabitation Single	5 (31.3) 11 (64.7)	10 (35.3) 6 (66.7)	0.077
Education High school University	11 (68.8) 5 (31.3%)	3 (18.8) 13 (81.3)	0.006
Employment Employed Retired	7 (43.8) 9 (56.3)	8 (50.0) 8 (50.0)	0.723
Residence Rural Urban/Suburb	7 (43.8) 9 (56.3)	5 (31.3) 11 (68.8)	0.465
Currently smoking No Yes	9 (56.3) 7 (43.8)	1 (6.3) 15 (93.8)	0.002
Alcohol use Yes No	11 (68.8) 5 (31.3)	6 (37.5) 10 (62.5)	0.156
Concomitant diseases Cardiac failure ^a^ COPD ^b^ and asthma Diabetes ^c^ Thyroid	16 (100.0) 2 (12.5) 5 (31.3) 1 (6.3)	16 (100.0) 6 (37.5) 11 (68.8) 1 (6.3)	1.000 0.220 0.034 1.000
Number of co-morbidities ^d^ 1 2 3 4	6 (37.5) 5 (31.3) 4 (25.1) 1 (6.3)	3 (18.8) 3 (18.8) 4 (25.1) 6 (37.5)	0.045

^a^ Arterial hypertension, ischemic heart disease, and cardiac insufficiency; ^b^ chronic obstructive pulmonary disease; ^c^ all types of diabetes; ^d^ cardiac insufficiency, COPD, asthma, and diabetes.

**Table 5 ijerph-21-00605-t005:** Univariate analysis of major improvements in the observed variables compared to minor improvements (*n* = 32).

Variable	Minor Improvement *n* (%)	Major Improvement *n* (%)	OR (95% CI)	*p* Value
Sex Male Female	13 (81.3) 3 (18.7)	15 (93.8) 1 (6.3)	0.289 (0.027–3.127)	0.307
Age (mean ± SD)	62.06 ± 7.9	59.06 ± 6.5	0.941 (0.849–1.043)	0.247
Marital status Married/cohabitation Single	5 (31.3) 11 (64.7)	10 (35.3) 6 (66.7)	3667 (0.849–15.844)	0.082
Education High school University	11 (68.8) 5 (31.3%)	3 (18.8) 13 (81.3)	5010 (1.397–17.971)	0.013
Employment Employed Retired	7 (43.8) 9 (56.3)	8 (50.0) 8 (50.0)	0.778 (0.193–3.127)	0.723
Residence Rural Urban/Suburb	7 (43.8) 9 (56.3)	5 (31.3) 11 (68.8)	1.711 (0.403–7.271)	0.467
Currently smoking No Yes	9 (56.3) 7 (43.8)	1 (6.3) 15 (93.8)	9.533 (1.847–49.204)	0.007
Alcohol use Yes No	11 (68.8) 5 (31.3)	6 (37.5) 10 (62.5)	1.941 (0.952–3.957)	0.068
Concomitant diseases Cardiac insufficiency ^a^ COPD ^b^ and asthma Diabetes ^c^ Thyroid	16 (100.0) 2 (12.5) 5 (31.3) 1 (6.3)	16 (100.0) 6 (37.5) 11 (68.8) 1 (6.3)	4.200 (0.698–25.264) 4.840 (1.085–21.583)	1.000 0.117 0.039 1.000
Number of co-morbidities ^d^ 1 2 3 4	6 (37.5) 5 (31.3) 4 (25.1) 1 (6.3)	3 (18.8) 3 (18.8) 4 (25.1) 6 (37.5)	2.033 (1.007–4.105)	0.045

^a^ Arterial hypertension, ischemic heart disease, and cardiac insufficiency; ^b^ chronic obstructive pulmonary disease; ^c^ all types of diabetes; ^d^ cardiac insufficiency, COPD, asthma, and diabetes.

## Data Availability

The raw data supporting the conclusions of this article will be made available by the authors, without undue reservation.

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
