# Peer review of "Smart Anklet Use to Measure Vascular Health Benefits of Preventive Intervention in a Nature-Based Environment—A Pilot Study"

_ijerph, 2024, doi:10.3390/ijerph21050605_

Round 1
Reviewer 1 Report
Comments and Suggestions for Authors
This study provides good information regarding the potential positive effect of physical activity on peripheral arterial occlusive disease. However, there are two issues that need to be tackled by the authors before the study may be publishable.
The first issue regards the absence of a control group or of a waiting-list control group or of a comparison group. The inclusion of one, or more, of such groups would strengthen the study significantly. As it stands, one cannot conclude that physical activity is a better or worse “treatment” for PAOD. The absence of a control group renders the intervention and the corresponding results rather weak. I would suggest that the authors state this explicitly in the limitations section and provide suggestions for rectifying this issue in future studies.
The second issue refers to the precision of reporting. In various places in the manuscript, there are not sufficient details for the reader to have a detailed picture or the reporting is unclear. I provide three examples: a. The authors state that they took three measures of wave amplitude the day before and after the intervention. They should also state how they combined the three measures. b. In the discussion section, the authors state that “Major improvements of the peripheral pulse parameters among our study participants, during the physical activity in nature- 228 based environment, were associated with current smoking.” It is not clear from this wording, whether the improvements were for all patients regardless of their smoking status, or whether the improvement was larger for those who smoked in comparison to those that did not smoke. c. In the study design section it is stated that “It consisted of a prospective cohort and two cross-sectional studies. The first cross-sectional study was conducted before nature-based intervention while the second was conducted six months later.” It is unclear to me as a reader whether one or two studies were implemented.
In light of the above examples, I suggest that: a. the authors should carefully restructure the Methods section. Please add a new sub-section “Instrument(s)” to describe the “smart anklet apparatus”. Please add a new sub-section “Measures” to describe measures and measurement in detail (please also refer to my comment above regarding the three measures of wave amplitude”. You may also move there the 1st paragraph of the Statistical Analysis section. b. Please examine carefully and edit the whole manuscript for clarity and precision.
Additional comments:
Were the participants of the study under medication for PAOD?
In the abstract you refer to PAOD for the first time. Please spell out there what PAOD stands for.
Comments on the Quality of English LanguageThe language needs careful editing
Author Response
Corresponding Author:
Maja Stosic, MD, PhD, Assistant Professor
Faculty for Health and Business Studies, Singidunum University, Valjevo, Serbia
Institute of Public Health of Serbia “Dr Milan Jovanovic Batut”, Belgrade, Serbia
Tel/Fax: +381641278571
E-mail: maja_stosic@batut.org.rs
majavstosic@gmail.com
Belgrade, April 23, 2024
To The Editor, International Journal of Environmental Research and Public Health
Dear Editor,
We are pleased to submit our revised manuscript “Smart anklet" to measure vascular health benefit of preventive intervention in environment of nature – a pilot study” by Stosic M. et al. for publication in the respectful journal such as International Journal of Environmental Research and Public Health.
We are grateful for the comments of the reviewer and attentive review to improve our manuscript. We have replied point-by-point to all issues raised by the reviewer. All changes in the manuscript are marked using “Track Changes” function.
Thank you for opportunity to revise our manuscript and we hope it will be suitable for publication in the International Journal of Environmental Research and Public Health.
Sincerely,
Maja Stosic
Responses to Reviewer 1
Comments and Suggestions for Authors
This study provides good information regarding the potential positive effect of physical activity on peripheral arterial occlusive disease. However, there are two issues that need to be tackled by the authors before the study may be publishable.
- The first issue regards the absence of a control group or of a waiting-list control group or of a comparison group. The inclusion of one, or more, of such groups would strengthen the study significantly. As it stands, one cannot conclude that physical activity is a better or worse “treatment” for PAOD. The absence of a control group renders the intervention and the corresponding results rather weak. I would suggest that the authors state this explicitly in the limitations section and provide suggestions for rectifying this issue in future studies.
ANSWER 1: We thank for the comment and fully agree that the control group measurement would controbuted significantly to the strenght of the study. Unfortunately, due to cost limitations and limitations in the number of study participants, they could not be randomly assigned. Therefore, pretest-posttest design, as quasi-experimental, was used within the pilot with the aim to assess initial associations between an independent and dependent variables that should be futher explored in future research. As recommended, in that sense, we revised the limitations section as follows:
It has several limitations. The main limitation of the study is absence of the control group. Due to limitation in cost and time, and inability of certain number of patients with PAOD to participate in the study due to objective inability to perform continuous physical activity, we included a limited number of respondents. It resulted in non-equal distribution of patients between the age groups. In addition, we measured only several peripheral pulse characteristics. Therefore, a larger sample size is needed to explore further associations and increase the strength of the study. Despite limitations, our study results provided initial associations between independent and dependent variable and possibilities of use of smart devices for measurement of peripheral pulse characteristics. This findings should be further explored in future research.
- The second issue refers to the precision of reporting. In various places in the manuscript, there are not sufficient details for the reader to have a detailed picture or the reporting is unclear. I provide three examples:
2a. The authors state that they took three measures of wave amplitude the day before and after the intervention. They should also state how they combined the three measures.
ANSWER 2a. Thank you for the comment. The average values were calculated for the initial assessment and compared to average values of the measurements within the second assessment. This statement was added to the sub-subsection Measures of the Methods section (lines 101-102).
2b. In the discussion section, the authors state that “Major improvements of the peripheral pulse parameters among our study participants, during the physical activity in nature- 228 based environment, were associated with current smoking.” It is not clear from this wording, whether the improvements were for all patients regardless of their smoking status, or whether the improvement was larger for those who smoked in comparison to those that did not smoke.
ANSWER 2b: Thank you for the comment. The improvement was larger for those who smoked in comparison to those that did not smoke. As it was not clear for the reviewer, we added this sentence to clarify (lines 282-283).
2c. In the study design section it is stated that “It consisted of a prospective cohort and two cross-sectional studies. The first cross-section of the study was conducted before nature-based intervention while the second was conducted six months later.” It is unclear to me as a reader whether one or two studies were implemented.
ANSWER 2c: Thank you for the comment, highly appreciated. A panel study is a type of longitudinal research where data is collected from the same individuals (cohort), known as a panel, repeatedly over a period of time, to make inferences about trends, patterns, and causal relationships in that population. Panel studies involve sampling a cross-section of individuals at specific intervals for an extended period of time.
We revised the study design section and provided clarification as follows: We performed a longitudinal panel study from 1 January 2022 to 31 December 2022 at the vascular surgeons’ hospital “Dobb in Valjevo, Serbia. The first cross-section of the study participants was conducted before nature-based intervention while the second was conducted six months later.
In addition, we added clarification within the Abstract.
In light of the above examples, I suggest that: a. the authors should carefully restructure the Methods section. Please add a new sub-section “Instrument(s)” to describe the “smart anklet apparatus”. Please add a new sub-section “Measures” to describe measures and measurement in detail (please also refer to my comment above regarding the three measures of wave amplitude”. You may also move there the 1st paragraph of the Statistical Analysis section. b. Please examine carefully and edit the whole manuscript for clarity and precision.
We appreciate the comment. As suggested, we reordered the subsections within the Materials and methods section to increase clarity and precision. Within the Data collection subsection, we introduced new sub-subsections “Measures“ and “Instrument(s)“ where we adressed the issues one by one as follows (lines 79-129):
2.3. Data collection
2.3.1 Measures
The peripheral pulse characteristics for the study purposes were measured by the specially designed device named “smart anklet” (picture 1).
After the initial assessment and measurement performed by three measurements (in the morning, in the afternoon and in the evening), patients were exposed six months to moderate-intensity physical activity (MPA) in nature-based environment. They practiced 150 to 300 minutes of walking 6 kilometers (km) per hour (h) and cy-cling activities (16-20 km/h) weekly as recommended for the patients with chronic conditions and those living with disability.
Second assessment was performed the day after six months of physical activity, by three measurements the same way as initial assessment.
Variables monitored before and after preventive intervention were:
- pain when walking self-reported by the participants
- characteristics from the measurements by the “smart anklet” device: claudication distance defined as the distance after which the patient is forced to stop because of severe pain and muscle cramps, wave amplitude – damped, wave amplitude – delayed and wave amplitude – reduced.
The average value was calculated for the initial assessment and compared to the average value of the measurements within the second assessment.
In addition, patients were classified by the Fontaine stage before and after the preventive intervention. The Fontaine classification describes four stages in clinical presentation of lower extremity arterial disease:
Stage I – patients who are asymptomatic for most of the time, but in whom a careful history may reveal non-specific, subtle symptoms, such as paresthesia. Physical examination may reveal cold extremities, reduced peripheral pulse or murmurs in the peripheral arteries.
Stage II – Intermittent claudication. Patients usually have a constant distance at which the pain appears:
Stage IIa – Intermittent claudication after more than 200 m of walking.
Stage IIb – Intermittent claudication after less than 200 m of walking.
Stage III – Rest pain. Rest pain appears especially during the night when the legs are raised up on to the bed, which diminishes the gravitational effect present by day; also, during the night, the lack of sensory stimuli allows patients to focus on their legs.
Stage IV – Ischaemic ulcers or gangrene (which may be dry or humid) (13)
To assess the outcomes of the preventive interventions, the improvement scores were calculated for the following variables: pain when walking, claudication distance defined as the distance after which the patient is forced to stop because of severe pain and muscle cramps, wave amplitude – damped, wave amplitude – delayed and wave amplitude – reduced. If the improvement is presented, the variable is marked with 1, if not with 0. The highest score was five and the lowest was 0.
Based on the median distribution of scores, the improvements were classified as minor and major. The improvement is classified as minor if 1 or 2 variables achieved improvements, while major if 3 or more variables were improved during the study period.
Results were presented as frequency (percent), median (range) and mean ± stand-ard deviation (SD). Univariate logistic regression was used as the method for analyz-ing binary outcomes and potential predictors. Due to small number of the study par-ticipants, it was not feasible to design multivariate logistic regression (MLRA) model. All p-values less than 0.05 were considered statistically significant. Statistical data analysis was performed using IBM SPSS Statistics 22 (IBM Corporation, Armonk, NY, USA).
Picture 1. Smart anklet device.
2.3.2 Instrument
Additional comments:
Were the participants of the study under medication for PAOD?
ANSWER 3: Thank you for the question? The patients included in the study used regularly their medications. The study had no influence on therapy. This statement was added in the Materials and methods section (line 77-78).
In the abstract you refer to PAOD for the first time. Please spell out there what PAOD stands for.
ANSWER 4: Thank you for the comment. We explained the abbreviation PAOD for peripheral arterial occlusive disease, before its first appearance in the abstract.
Comments on the Quality of English Language
The language needs careful editing
ANSWER 5: We appreciate the comment. We revised the quality on english language throughout the text by the professional translator.

Reviewer 2 Report
Comments and Suggestions for Authors
1. Table 1 is not necessary to supoort your aim. it may comes after table 4 appropriately.
2.please put individual p value in each tables in order to able to see significances. ex) was p value 0.001 cames out between before and after or yes or nor in the walking pain of variable? have you done age and sex match on data?
3. is there report any causation of claudication itself by smart anklet?
4. would you provide the reason, it seems non significant between in residence whether rural or urban. since the authors mentioned that positive associations between exposure to nature and physical activity in the introduction as well as in the title.
Author Response
Corresponding Author:
Maja Stosic, MD, PhD, Assistant Professor
Faculty for Health and Business Studies, Singidunum University, Valjevo, Serbia
Institute of Public Health of Serbia “Dr Milan Jovanovic Batut”, Belgrade, Serbia
Tel/Fax: +381641278571
E-mail: maja_stosic@batut.org.rs
majavstosic@gmail.com
Belgrade, April 23, 2024
To The Editor, International Journal of Environmental Research and Public Health
Dear Editor,
We are pleased to submit our revised manuscript “Smart anklet" to measure vascular health benefit of preventive intervention in environment of nature – a pilot study” by Stosic M. et al. for publication in the respectful journal such as International Journal of Environmental Research and Public Health.
We are grateful for the comments of the reviewer and attentive review to improve our manuscript. We have replied point-by-point to all issues raised by the reviewer. All changes in the manuscript are marked using “Track Changes” function.
Thank you for opportunity to revise our manuscript and we hope it will be suitable for publication in the International Journal of Environmental Research and Public Health.
Sincerely,
Maja Stosic
Responses to Reviewer 2
Comments and Suggestions for Authors
- Table 1 is not necessary to support your aim. it may comes after table 4 appropriately.
ANSWER 1: Thank you for the comment. We deleted table 1 and revised the text of the results to cover data from the table as follows (line 232-236): Less than a half were smokers (43.8%), while more than a half of study participants used alcohol (53.1%) at the time of the study. One fourth of them had chronic obstructive pulmonary disease (COPD) and asthma, one half had diabetes and 6.3% thyroid disease. More than fourth of them (28.1%) had one co-morbidity, 25% had two, 25% three and 21.9% four co-morbidities
- please put individual p value in each tables in order to able to see significances. ex) was p value 0.001 cames out between before and after or yes or nor in the walking pain of variable? have you done age and sex match on data?
ANSWER 2: Thank you for the comment. The values are not adjusted nor matched. The p values are from the chi square tests in tables 1, 2 and 4, while for table 5, the p values were taken from the logistic regression analysis. All p values comes out from the differences before and after the intervention and are the same regardless of the cathegory used as a reference.
- is there report any causation of claudication itself by smart anklet?
ANSWER 3: Thank you for the comment. Unfortunately, this is the initial measurement with this device, so far it cannot identify any causation of claudication itself by smart anklet, since it can be caused by different factors. It is able to assess association between the physical activity and claudication distance. Once the claudication occured, the patient stopped walking. We mesured whether the psysical activity during six months period affected the claudication distance.
- would you provide the reason, it seems non significant between in residence whether rural or urban. since the authors mentioned that positive associations between exposure to nature and physical activity in the introduction as well as in the title.
ANSWER 4: Thank you for the comment. The first reason might be the low number of patients and therefore detailed disaggregation was not possible. In addition, the study participants are patients with predominantly severe forms of peripheral arterial occlusive disease (87,5% at Fontaine stage IIb), not able for prolonged walking regardless of the residence. Their disease characteristics is the crucial determinant of physical activity, not residency in this case and therefore the results are assumed as conceptually appropriate.
